# Incidence of Asthma, Atopic Dermatitis, and Allergic Rhinitis in Korean Adults before and during the COVID-19 Pandemic Using Data from the Korea National Health and Nutrition Examination Survey

**DOI:** 10.3390/ijerph192114274

**Published:** 2022-11-01

**Authors:** Hyo Geun Choi, So Young Kim, Yeon-Hee Joo, Hyun-Jin Cho, Sang-Wook Kim, Yung Jin Jeon

**Affiliations:** 1Departments of Otorhinolaryngology-Head & Neck Surgery, Hallym University College of Medicine, Anyang 14068, Korea; 2Department of Otorhinolaryngology-Head & Neck Surgery, CHA Bundang Medical Center, CHA University, Seongnam 13496, Korea; 3Department of Otorhinolaryngology, Gyeongsang National University Changwon Hospital, Changwon 51472, Korea; 4Institute of Health Sciences, Gyeongsang National University, Jinju 52727, Korea; 5Department of Otorhinolaryngology, Gyeongsang National University Hospital, Jinju 52727, Korea

**Keywords:** COVID-19, incidence, asthma, atopic dermatitis, and allergic rhinitis

## Abstract

The prevalence of allergic diseases has been increasing globally prior to COVID-19. The pandemic resulted in changes in lifestyle and personal habits such as universal mask-wearing and social distancing. However, there is insufficient information on the impact of the COVID-19 pandemic on the prevalence of allergic conditions such as asthma, atopic dermatitis, and allergic rhinitis. We analyzed the incidence rate for self-reported and doctor-diagnosed cases of allergic diseases of asthma, atopic dermatitis, and allergic rhinitis. A total of 15,469 subjects were registered from a national cohort dataset of the National Health and Nutrition Examination Survey. Using multiple logistic regression analysis, we calculated the adjusted odds ratio (OR) for each disease in 2020 compared to 2019. Subgroup analyses were performed according to age and sex. There were no statistically significant differences between the incidence of doctor-diagnosed and current allergic diseases in 2019 and 2020 (asthma, *p* = 0.667 and *p* = 0.268; atopic dermatitis, *p* = 0.268 and *p* = 0.973; allergic rhinitis, *p* = 0.691 and *p* = 0.942, respectively), and subgroup analysis showed consistent results. Among the Korean population from 2019 to 2020, the incidence of the allergic diseases asthma, atopic dermatitis, and allergic rhinitis did not decrease as expected.

## 1. Introduction

A new strain of human coronavirus emerged in 2019 and caused an epidemic and has been designated by the International Committee on Taxonomy of Viruses (ICTV) as Severe Acute Respiratory Syndrome Coronavirus 2 (SARS-CoV-2) [1]. The disease was named “Coronavirus Infectious Disease 2019 (COVID-19)” by the World Health Organization (WHO). Since the first outbreak in Wuhan, China, in December 2019, COVID-19 spread aggressively around the world [2]. On 20 January 2020, the Republic of Korea reported its first case of COVID-19. The Korea Centers for Disease Control and Prevention (KCDC) reported a rapid increase in continuous transmission of COVID-19 on 19 February 2020 [3]. The WHO proclaimed a COVID-19 pandemic on 11 March 2020. Lifestyles changed dramatically to prevent the spread of epidemic infections and have remained affected to the present [4]. Official directives such as increased hand washing, mask-wearing, social distancing, and working from home are not only affecting the dispersion of COVID-19, but also the spread of other transmittable diseases.

Atopic diseases tend to be an exaggerated immunoglobulin E-mediated immune response in response to the foreign allergen [5]. Patients with atopic traits usually present with one or more symptoms of the following disorders: asthma, atopic dermatitis, and allergic rhinitis. In recent decades, the prevalence of allergic diseases has been steadily increasing, and it currently affects about 20% of the population in developed countries [6,7]. Allergic diseases can develop at a relatively young age and reduce the quality of life [8]. Allergic diseases are common chronic and recurrent inflammatory diseases. Continuous management is required to treat allergic diseases, and the socioeconomic burden is increasing worldwide. In recent decades, the prevalence of allergic diseases has raised rapidly worldwide, especially in low- and middle-income countries [9,10]. However, evidence supporting such trends in asthma, atopic dermatitis, and allergic rhinitis in the Republic of Korea is controversial. Some previous studies have reported that the prevalence has increased, and others have reported that this prevalence has decreased [11,12,13,14]. In a recent study using large population data distributed by the National Health and Nutrition Examination Survey (KNHANES), the 10-year trend in asthma prevalence reported from 2008 to 2017 was stable [6].

Social restrictions including universal masking requirements, social distancing, and national lockdowns were intended to reduce the burden of COVID-19 mortality and morbidity. Additionally, these social limitations have been associated with a reduction in non–SARS-CoV-2 infections such as seasonal influenza, other respiratory infections, and foodborne infectious diseases [15,16,17,18]. Individuals had to adapt by changing their daily routines. The quarantine has disrupted lifestyle behaviors, including those associated with work, education, physical exercise, sleep, and food consumption [19,20,21,22]. To prevent viral spread, public places such as offices, schools, gyms, and restaurants have closed or have severely limited occupancy. However, there is insufficient information concerning the influence of the COVID-19 pandemic on the prevalence or severity of allergic diseases of asthma, atopic dermatitis, and allergic rhinitis. In this study, we examined and compared the epidemiologic characteristics and the proportions of ‘doctor-diagnosed’ and ‘currently diagnosed’ allergic diseases to evaluate the incidence of allergic diseases from 2019 (before COVID-19 appearance) and 2020 (during the COVID-19 epidemic).

## 2. Materials and Methods

### 2.1. Study Population and Data Collection

The KNHANES (https://knhanes.kdca.go.kr/knhanes/main.do, accessed on 31 August 2022) was conducted by the KCDC in the interest of public welfare. Institutional Review Board (IRB) review and authorization requirements for the present study were waived by the KCDC. This study meets the standards described in paragraph 2 and subparagraph 1 of the Bioethics and Safety Act and paragraph 1, subparagraph 1 of the Enforcement Rule of the Bioethics and Safety articles. All KNHANES data analyses were performed under the guidelines provided by the IRB of the KCDC. The feasibility, understanding, and reliability of each questionnaire were surveyed by the KCDC to authenticate the applicability of the questionnaires.

These repeated cross-sections used the KNHANES database from the national medical insurance provider. Database-associated statistical procedures were based on constructed sampling and corrected weighted values. The database was assembled by the KCDC. A panel chooses a yearly sample of 25 households from 192 account districts to create a database that reflects the entire Korean population. Statisticians who conducted post-stratification analyses weighted the sample to reflect extreme values and non-response rates. Data from the eighth KNHANES performed between 2019 and 2020 were analyzed. Because the first positive case of COVID-19 in the Republic of Korea was identified on 20 January 2020, a comparison of data from 2019 and 2020 was deemed appropriate. This study did not follow enrolled participants from 2019 to 2020, but analyzed data from newly-extracted individuals each year.

The KNHANES website (https://knhanes.kdca.go.kr/knhanes/main.do, accessed on 31 August 2022) describes the details of the sampling procedures. Of the total of 15,469 participants (8110 in 2019; 7359 in 2020), exclusions from the study included participants younger than 19 (n = 2730), those missing body mass index (BMI) data (n = 653), those missing income data (n = 59), and those without a sleep time record (n = 13). Remaining for inclusion in the study were 12,014 participants (6218 from 2019 and 5796 from 2020) older than 19 years (Figure 1). We investigated the prevalence of asthma, atopic dermatitis, and allergic rhinitis between 2019 and 2020 based on self-reporting.

### 2.2. Survey

#### 2.2.1. Exposure

In 2019 and 2020 surveys, we selected adult participants meeting inclusion and exclusion criteria who represented the entire population of the Republic of Korea. We did not follow the 2019 participants; 2020 participants were newly-selected from the entire Korean population.

#### 2.2.2. Outcome

Data were acquired on medical histories of asthma, atopic dermatitis, and allergic rhinitis. The relevant survey questions were: “Have you been diagnosed with asthma by a doctor within 12 months?”. If yes, the condition was classified as ‘doctor-diagnosed asthma.’ Additionally, the participants were asked “Are you currently being treated for asthma?” If yes, the condition was classified as ‘current asthma.’ Atopic dermatitis and allergic rhinitis were surveyed on the same questionnaire [23].

#### 2.2.3. Covariate

Income was recalculated as the distribution of total household income value by the square root of the total number of family members [24]. We classified employment status as either unemployed or employed. We divided educational status as college or higher, high school, junior high school, elementary school or lower, or unknown. We surveyed and classified house type as detached house, condominium, townhouse, or others. We classified marital status as married, unmarried, or unknown. We evaluated BMI (kg/m^2^) using height and weight. Histories of smoking and alcohol consumption were assessed. Sleep time was calculated as 5/7 time on the weekday plus 2/7 time on the weekend [25].

### 2.3. Statistical Analysis

General 2019 and 2020 population characteristics were evaluated using linear regression analysis with complex sampling. The Chi-square test with Rao-Scott correction was performed on population data to allow for manipulation of weighted values.

We calculated the crude and adjusted odds ratios (ORs) for asthma, allergic rhinitis, and atopic dermatitis in 2020 compared to 2019 through multiple logistic regression analysis with complex sampling. We adjusted for confounding continuous factors of age, income, BMI, and sleep time. The adjusted categorical variables were employment status, education history, house type, marital status, smoking history and alcohol consumption.

We designed the model to allow analysis by age and sex subgroups. We performed two-tailed analyses, and *p*-values lower than 0.05 were regarded as indicating significance; 95% confidence intervals (CIs) were also analyzed. We applied the weights recommended by the KNHANES and presented all results as weighted values. We analyzed the data using SPSS ver. 25.0 (IBM, Armonk, NY, USA).

## 3. Results

### 3.1. General Differences of Demographic Data and Incidence Rates of ‘Doctor-Diagnosed’ and ‘Current’ Allergic Diseases in the Korean Adult Population between 2019 and 2020

This study included 6218 participants in 2019 and 5796 participants in 2020. Table 1 presents the baseline characteristics and comorbidities of the included samples. The average age of all participants was 51.95 ± 17.1 years. In 2020, the population BMI was higher (24.2 ± 3.8 kg/cm^2^) than that in 2019 (23.9 ± 3.6 kg/cm^2^, *p*-value < 0.001). There were no significant differences in doctor-diagnosed conditions within the applicable 12 months for asthma (2.9% [178/6218] vs. 3.0% [176/5796], *p*-value = 0.819), atopic dermatitis (3.0% [189/6218] vs. 3.5% [204/5796], *p*-value = 0.197), and allergic rhinitis (14.1% [878/6218] vs. 14.6% [847/5796], *p*-value = 0.762) between 2019 and 2020.

### 3.2. Adjusted Odds Ratio of ‘Doctor Diagnosed’ and ‘Current’ Allergic Diseases in 2020 Compared to 2019 in Korean Adult Population

The adjusted odds ratio (OR) of doctor-diagnosed and current asthma, atopic dermatitis, and allergic rhinitis were calculated for the 2019 and 2020 participants. The odds ratio was adjusted for age, sex, income, employment status, educational history, house type, marital status, BMI, smoking status, alcohol consumption, and sleep duration. No significant differences were found in the adjusted ORs for doctor-diagnosed and current asthma (0.94 [95% CI = 1.16–1.94, *p* = 0.667] and 1.15 [95% CI = 0.90–1.48, *p* = 0.268], Table 2), atopic dermatitis (1.15 [95% CI = 0.90–1.64, *p* = 0.268] and 1.00 [95% CI = 0.67–1.48, *p* = 0.989], Table 3), and allergic rhinitis (1.03 [95% CI = 0.90–1.18, *p* = 0.691] and 1.00 [95% CI = 0.86–1.15, *p* = 0.942], Table 4) in 2020 compared to that in 2019. We performed subgroup analyses by sex and economic level. The results consistently demonstrated no difference according to sex or economic level for asthma, atopic dermatitis, and allergic rhinitis.

## 4. Discussion

In this study, we showed that the reported rates of doctor-diagnosed and current allergic diseases of asthma, atopic dermatitis, and allergic rhinitis among Korean populations in 2020 remained unchanged from 2019. BMI has increased since the beginning of the COVID-19 pandemic, correlated with a decrease in physical activity due to viral spread prevention methods [20]. To the best of our knowledge, this is the first study to focus on the incidence of newly-diagnosed allergic diseases before and during the COVID-19 pandemic. In this study, we adjusted for several potential confounding factors. The age-stratified nationwide repeated cross-sections of KNHANES data over the 10 years from 2008 to 2017 revealed a stable 10-year trend for incidence of asthma; an increased incidence of allergic rhinitis during this period, especially in adults (from 19 to 59 years old) and the elderly (older than 60 years) [6]. In 2020, the incidences of newly-diagnosed asthma, atopic dermatitis, and allergic rhinitis are likely related to pandemic-associated lifestyle changes resulting from government regulations to control the COVID-19 pandemic. These changes had not been necessary in the previous year.

We previously reported that the incidence of AR and asthma in the Korean adolescent 7th through 12th grade population decreased significantly from 2019 to 2020, but atopic dermatitis incidence remained constant [26]. In the current study, the results for the incidence of allergic diseases were not consistent with our hypothesis. Considering the extensive efforts for holistic prevention of COVID-19 in 2020 compared to 2019, we supposed that new diagnoses of allergic diseases would decrease like those of annual respiratory viral infectious diseases such as influenza, parainfluenza, and respiratory syncytial virus. These efforts, particularly of universal mask-wearing policies, were expected to decrease allergen exposure. Universal mask-wearing did, indeed, significantly reduce overall allergic rhinitis symptoms when N95 masks (12.6%, *p* = 0.027) and surgical masks (13.0%, *p* = 0.039) were used, but there were no significant improvements in mild or persistent allergic rhinitis [27]. Significantly, during the COVID-19 pandemic, patients with symptoms of allergic diseases have had fewer opportunities to obtain prescriptions due to reduced clinical capacities [28]. The overall reduction in hospital visits during the COVID-19 pandemic does not simply coincide with a reduced incidence of allergic diseases. This includes changing caregiver behavior for patients who are more likely to avoid visiting the hospital because of fear of transmitting SARS-CoV-2 and stopping health care [29,30].

During the COVID-19 pandemic, non-pharmacological measures such as wearing a mask, washing hands frequently, and improving indoor ventilation improved hygiene and self-care, including high medication adherence rates [30,31]. In accordance with hygiene awareness, non-pharmacological interventions reduced exposure to outdoor exposure to pollen allergens and air pollution played a protective role, and alleviated allergic reactions. Consequently, the average daily asthma emergency treatment rate decreased by 21% to compared to 76% before the COVID-19 pandemic [32,33,34], and the number of children hospitalized for asthma decreased [31,35,36]. Reducing exposure to common respiratory viruses, which is the major trigger for asthma exacerbation, through non-pharmacological interventions and social distancing may be the most plausible explanation, according to some studies [30,33].

We expected an overall decrease in the incidence of allergic diseases, such as reduced asthma exacerbations, during the COVID-19 pandemic. Interestingly, the results differed from what was initially expected. The trend of allergen positivity has changed following non-pharmaceutical interventions in the spread of SARS-CoV-2 infection during the COVID-19 pandemic [33]. Evidence of the effect of avoidance therapy for house dust mites and pet allergens according to the 2008 ARIA guidelines is insufficient [37]. Moreover, behavior changes related to quarantine increased exposure to indoor allergens including house dust mites, mold, animal dander, and cockroaches [4,38,39]. Increases in allergic disease incidence may have been associated with these exposures. A negative effect of COVID-19 lockdown was that, in 2020, up to 38.4% of allergic rhinitis patients reported a worsening of symptoms [38]. Those with house dust mite allergies reported higher sinonasal outcome test (SNOT-22) scores and increased use of systemic antihistamines and nasal decongestants compared to 2019 [40]. In atopic dermatitis, genetic and environmental factors determine susceptibility. Most adult-onset atopic dermatitis is related to sensitivity to indoor aeroallergens such as animal dander and house dust mites [8,41]. It is likely that the increased exposure to indoor allergens counteracted the decreased exposure to outdoor viruses and outdoor air pollutants, and the prevalence of allergic diseases did not decrease as expected due to the increased exposure to indoor allergens.

Studies have demonstrated that the COVID-19 pandemic has been associated with unhealthy eating behaviors and reduced physical activity. These behaviors and increased sedentary habits have led to dysglycemia, higher metabolic risk, and weight gain [42,43,44]. During the COVID-19 pandemic, home confinement, social distancing, and school suspension have been associated with reduced sleep duration and impaired patterns and quality of sleep [45,46]. The initiation and maintenance of the sleep-wake cycle are described by a two-phase model in which circadian and homeostatic factors continuously interact to initiate and maintain sleep [47]. A circadian factor increases the likelihood of sleep initiation. A homeostatic factor affects sleep pressure and is proportional to awake time; more awake time during the day results in lower sleep pressure. Home confinement can disrupt the circadian rhythm by shifting to later waking and bedtime and increasing daytime napping. These contributed to a reduction in sleep pressure during the COVID-19 pandemic [48,49]. In the current study, BMI increased significantly in 2020, but sleep duration did not differ from 2019. Data on sleep time in the current study were the product of subjective questionnaires lacking objective results for sleep quality, such as nocturnal polysomnographic data. Further investigation is needed to confirm sleep disturbances during the COVID-19 pandemic.

The strengths of the present study are the population-based design and the large size of the sample. Moreover, the design of the present study allowed for the evaluation of the incidence of various allergic diseases. This incidence is expected to be accurate as the rate is not only based on self-reports, but also on physician diagnoses. The established clinical guidelines improve diagnostic accuracy in the Korean population. We acknowledge that there are some limitations related to this study. This study evaluated whether the frequency of allergic diseases changed according to changes in personal lifestyle and habits, such as wearing a universal mask during the COVID-19 pandemic. The target outcomes were based on self-reported doctor’s diagnoses, not symptom-based ones. There will inevitably be missed diagnoses. Moreover, our results could not fully demonstrate an association between the severity of allergic diseases and lifestyle changes during the COVID-19 pandemic. This was due to the lack of information regarding the inherent weaknesses of the database. Because we depend on the survey items of KNHANES data, the lack of information on the etiology of allergic diseases such as the timing of exposure to symbiotic microbiota, nutrition, and housing characteristics may limit our ability to support our hypothesis [50,51,52]. Further studies are needed to confirm the presence of these factors on prevalence. The duration and degree of allergic diseases and the history of medications, such as oral or topical corticosteroids, were not analyzed in this study. While we adjusted for a variety of influencing factors, we could not eliminate all of the unmeasured or residual predisposing factors affecting the incidence of allergic diseases. Future studies should adjust for additional identified confounding factors. Since the national health service’s investigation period is from January to December, there may be limitations depending on the investigation. The spread of SARS-CoV-2 in the first quarter of 2020 has made mask-wearing compulsory in most areas, and many businesses have switched to telecommuting and postponed school attendance. The mandatory use of masks at the government level, which gives penalties and administers a fine for not wearing a mask, began on 13 October 2020. Although there may be a transitional period in social distancing, including mask-wearing, there was a limitation in setting the year for comparative analysis because we had to analyze using only the data during limited period provided by KNHANES. Further studies are needed to confirm changes in allergic disease prevalence through comparisons from different years. Despite these limitations, the results of the current study are ambiguous but interesting as objective evidence of the lack of correlation between the decreased incidence of allergic diseases during the COVID-19 pandemic.

## 5. Conclusions

Reports of asthma, atopic dermatitis, and allergic rhinitis did not change during the COVID-19 pandemic despite social restrictions and preventive measures such as universal masking. Further investigations are required with regard to the relationship between the incidence of allergic diseases and lifestyle changes to limit the spread of COVID-19.

## Figures and Tables

**Figure 1 ijerph-19-14274-f001:**
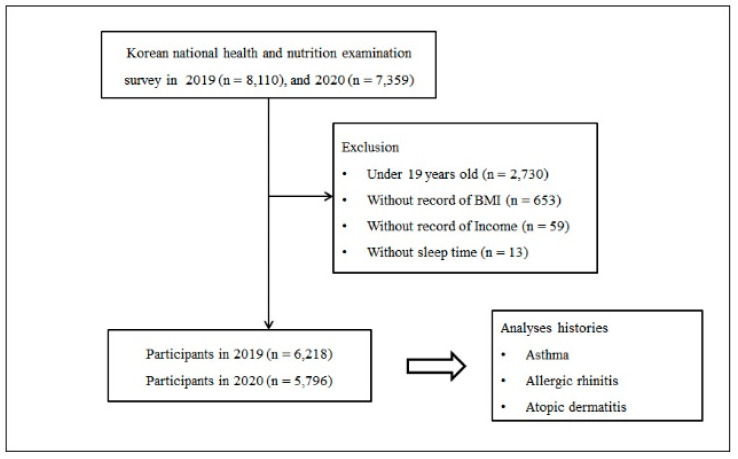
A schematic illustration of the participant selection process. Of 15,469 participants, data from 6218 in 2019 and 5796 in 2020 were compared. Abbreviations: BMI, body mass index.

**Table 1 ijerph-19-14274-t001:** General characteristics of participants.

Characteristics		Year		*p*-Value *
		2019	2020	
Age (yrs, mean, SD)		51.8 (16.9)	52.1 (17.3)	0.817
Age groups (yrs, n, %)				0.901
	19–39	1671 (26.9)	1556 (26.9)	
	40–59	2288 (36.8)	2021 (34.9)	
	≥60	2259 (36.3)	2219 (38.3)	
Sex (n, %)				0.873
	Males	2765 (44.5)	2610 (45.0)	
	Females	3453 (55.5)	3186 (55.0)	
Income (mean, SD)		3.2 (1.4)	3.2 (1.4)	0.588
Employment (n, %)				0.119
	Unemployed	2641 (42.5)	2570 (44.3)	
	Employed	3577 (57.5)	3226 (55.7)	
Educational status (n, %)				0.535
	Elementary school or under, unknown	1416 (22.8)	1347 (23.2)	
	Junior high school	557 (9.0)	517 (8.9)	
	High school	1964 (31.6)	1884 (32.5)	
	College or over	2281 (36.7)	2048 (35.3)	
House type (n, %)				0.683
	Detached house	1937 (31.2)	1888 (32.6)	
	Condominium	3392 (54.6)	3220 (55.6)	
	Raw houses	840 (13.5)	659 (11.4)	
	Others	49 (0.8)	29 (0.5)	
Marriage status (n, %)				0.082
	Married	4269 (68.7)	3764 (64.9)	
	Unmarried	883 (14.2)	866 (14.9)	
	Unknown	1066 (17.1)	1166 (20.1)	
Body mass index (mean, SD)		23.9 (3.6)	24.2 (3.8)	<0.001 *
Smoking status (n, %)				0.564
	Nonsmoker	3723 (59.9)	3499 (60.4)	
	Past smoker	1420 (22.8)	1333 (23.0)	
	Current smoker	1075 (17.3)	964 (16.6)	
Alcohol consumption (n, %)				0.133
	Non-consumer	2913 (46.9)	2853 (49.2)	
	1 to 5 times/mo	2005 (32.3)	1779 (30.7)	
	≥2 times/week	1300 (20.9)	1164 (20.1)	
Sleep duration (mean, SD)		6.9 (1.4)	7.0 (1.4)	0.187
Asthma (n, %)				
	Doctor diagnosed	178 (2.9)	176 (3.0)	0.819
	Current	95 (1.5)	111 (1.9)	0.198
Atopic dermatitis (n, %)				
	Doctor diagnosed	189 (3.0)	204 (3.5)	0.197
	Current	111 (1.8)	114 (2.0)	0.859
Allergic rhinitis (n, %)				
	Doctor diagnosed	878 (14.1)	847 (14.6)	0.762
	Current	718 (11.6)	668 (11.5)	0.887

Abbreviation; SD, standard deviation. * The general characteristics 2019, and 2020 were compared using linear regression analysis with complex sampling, and Chi-square test with Rao-Scott correction. Significance at *p* < 0.05.

**Table 2 ijerph-19-14274-t002:** Odds ratios (95% confidence intervals) for asthma (doctor diagnosed asthma, current asthma) in 2020 compared to 2019 with subgroup analyses according to age and sex.

Disease Subgroups			Odds Ratios for Asthma in 2020 Compared to 2019			
			Crude	*p*-Value	Adjusted †	*p*-Value *
Doctor diagnosed asthma						
	Total participants (n = 12,014)		0.97 (0.73–1.28)	0.819	0.94 (0.71–1.25)	0.667
	Age					
		19–39 years old (n = 3227)	1.03 (0.65–1.64)	0.891	0.97 (0.61–1.53)	0.880
		40–59 years old (n = 7536)	0.88 (0.50–1.54)	0.658	0.89 (0.51–1.55)	0.680
		≥60 years old (n = 4478)	0.95 (0.64–1.41)	0.799	0.97 (0.65–1.43)	0.865
	Sex					
		Males (n = 5375)	1.07 (0.71–1.60)	0.753	1.04 (0.70–1.57)	0.838
		Females (n = 6639)	0.88 (0.62–1.25)	0.484	0.87 (0.62–1.24)	0.446
Current asthma						
	Total participants (n = 12,014)		1.27 (0.88–1.85)	0.199	1.15 (0.90–1.48)	0.268
	Age					
		19–39 years old (n = 3227)	1.88 (0.85–4.13)	0.117	1.81 (0.78–4.17)	0.166
		40–59 years old (n = 7536)	1.33 (0.67–2.66)	0.416	1.45 (0.73–2.88)	0.287
		≥60 years old (n = 4478)	0.93 (0.59–1.44)	0.733	0.93 (0.60–1.44)	0.745
	Sex					
		Males (n = 5375)	1.47 (0.86–2.50)	0.160	1.43 (0.84–2.44)	0.186
		Females (n = 6639)	1.12 (0.71–1.76)	0.639	1.13 (0.71–1.81)	0.600

Abbreviation: BMI, body mass index. * Logistic regression, Significance at *p* < 0.05. † Adjusted for age, sex, income, employment, educational status, house type, marriage status, BMI, smoking status, alcohol consumption, and sleep duration.

**Table 3 ijerph-19-14274-t003:** Odds ratios (95% confidence intervals) for atopic dermatitis (doctor diagnosed atopic dermatitis, current atopic dermatitis) in 2020 compared to 2019 with subgroup analyses according to age and sex.

Disease Subgroups			Odds Ratios for Atopic Dermatitis in 2020 Compared to 2019			
			Crude	*p*-Value	Adjusted †	*p*-Value *
Doctor diagnosed atopic dermatitis						
	Total participants (n = 12,014)		1.18 (0.92–1.51)	0.199	1.15 (0.90–1.48)	0.268
	Age					
		19–39 years old (n = 3227)	1.27 (0.95–1.71)	0.111	1.22 (0.90–1.64)	0.205
		40–59 years old (n = 7536)	1.19 (0.67–2.12)	0.547	1.24 (0.70–2.18)	0.462
		≥60 years old (n = 4478)	0.73 (0.40–1.35)	0.318	0.77 (0.41–1.44)	0.409
	Sex					
		Males (n = 5375)	1.20 (0.84–1.72)	0.324	1.17 (0.81–1.70)	0.413
		Females (n = 6639)	1.16 (0.83–1.62)	0.390	1.12 (0.80–1.58)	0.505
Current atopic dermatitis						
	Total participants (n = 12,014)		1.03 (0.75–1.41)	0.859	1.00 (0.73–1.36)	0.973
	Age					
		19–39 years old (n = 3227)	1.08 (0.73–1.60)	0.712	1.00 (0.67–1.48)	0.989
		40–59 years old (n = 7536)	1.16 (0.59–2.31)	0.668	1.17 (0.60–2.29)	0.647
		≥60 years old (n = 4478)	0.69 (0.34–1.40)	0.307	0.72 (0.35–1.49)	0.369
	Sex					
		Males (n = 5375)	0.91 (0.59–1.41)	0.673	0.88 (0.57–1.36)	0.562
		Females (n = 6639)	1.15 (0.75–1.76)	0.516	1.09 (0.71–1.68)	0.694

Abbreviation: BMI, body mass index. * Logistic regression, Significance at *p* < 0.05. † Adjusted for age, sex, income, employment, educational status, house type, marriage status, BMI, smoking status, alcohol consumption, and sleep duration.

**Table 4 ijerph-19-14274-t004:** Odds ratios (95% confidence intervals) for allergic rhinitis (doctor diagnosed allergic rhinitis, current allergic rhinitis) in 2020 compared to 2019 with subgroup analyses according to age and sex.

Disease Subgroups			Odds Ratios for Allergic Rhinitis in 2020 Compared to 2019			
			Crude	*p*-Value	Adjusted †	*p*-Value *
Doctor diagnosed allergic rhinitis						
	Total participants (n = 12,014)		1.02 (0.89–1.18)	0.762	1.03 (0.90–1.18)	0.691
	Age					
		19–39 years old (n = 3227)	1.03 (0.85–1.25)	0.763	1.02 (0.84–1.24)	0.864
		40–59 years old (n = 7536)	1.06 (0.86–1.31)	0.574	1.08 (0.87–1.33)	0.493
		≥60 years old (n = 4478)	0.95 (0.72–1.26)	0.728	0.96 (0.73–1.26)	0.772
	Sex					
		Males (n = 5375)	1.02 (0.83–1.24)	0.880	1.03 (0.84–1.27)	0.750
		Females (n = 6639)	1.03 (0.87–1.22)	0.754	1.04 (0.88–1.22)	0.686
Current allergic rhinitis						
	Total participants (n = 12,014)		0.99 (0.85–1.15)	0.888	1.00 (0.86–1.15)	0.942
	Age					
		19–39 years old (n = 3227)	0.98 (0.80–1.20)	0.829	0.97 (0.79–1.19)	0.746
		40–59 years old (n = 7536)	1.05 (0.84–1.31)	0.680	1.06 (0.85–1.33)	0.594
		≥ 60 years old (n = 4478)	0.92 (0.67–1.27)	0.627	0.93 (0.67–1.27)	0.633
	Sex					
		Males (n = 5375)	0.92 (0.75–1.14)	0.441	0.94 (0.76–1.16)	0.547
		Females (n = 6639)	1.04 (0.87–1.26)	0.657	1.05 (0.88–1.26)	0.588

Abbreviation: BMI, body mass index. * Logistic regression, Significance at *p* < 0.05. † Adjusted for age, sex, income, employment, educational status, house type, marriage status, BMI, smoking status, alcohol consumption, and sleep duration.

## Data Availability

Not applicable.

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
