# Peer review of "Incidence of Asthma, Atopic Dermatitis, and Allergic Rhinitis in Korean Adults before and during the COVID-19 Pandemic Using Data from the Korea National Health and Nutrition Examination Survey"

_ijerph, 2022, doi:10.3390/ijerph192114274_

Round 1

Reviewer 1 Report (Previous Reviewer 1)

Thank you for revising you manuscript!

This manuscript is a resubmission of an earlier submission. The following is a list of the peer review reports and author responses from that submission.

Round 1

Reviewer 1 Report

The authors present analyses which are apparently supplementary analyses based on a health and nutritional survey study, the Korean equivalent of the well-known NHANES format. On line 79, the design is described as "cross-sectional cohort study" which seems a contradiction in itself; according to the description with new household samples being drawn in successive years, the design would rather correspond to repeated cross-sections.

Statistical analysis of the outcome variables was - appropriately - by multiple logistic regression (as described in the methods section). However, in the abstract, the authors write "Using linear regression analysis, we calculated the adjusted odds ratio (OR) for each disease" which is wrong.

The actual research question is whether atopic diseases have changed in frequency during the covid-19 epidemic; results are based on medical records which have been retrieved for the sampled individuals, comparing the years 2019 and 2020, the latter being the "Covid year". The target outcome was based on a self-reported doctor's diagnosis, not symptom-based.   

As result, the authors report non-significant variation of the OR to be diagnosed for the target outcome diseases in 2020, compared to 2019 as reference year. Thereby, their initial hypothesis "We expected a decreased incidence of allergic diseases during the COVID-19 pandemic due to ... changes in lifestyle and personal habits such as universal mask-wearing and social distancing" could not be confirmed.  

This result could be regarded as moderately interesting if, and only if, the above hypothesis would appear plausible. Of course it can always be claimed that there is "insufficient information concerning the influence of the COVID-19 pandemic on the prevalence or severity of allergic diseases of asthma, atopic dermatitis, and allergic rhinitis" as written in the introduction. Nevertheless, such lack is not a sufficient motivation for the analyses presented. Unfortunately, the authors made no attempt to relate to current concepts of the aetiology of atopic allergies as related to lifestyle factors (e.g. nutrition, housing, early exposure to a "rural" microbiome and others) - else they would perhaps have been aware of the limited sense of their hypothesis.

Author Response

Dear Editor,

We would like to thank the reviewers for their constructive criticism, which has helped us improve the manuscript. We have carefully and thoroughly addressed all the reviewers’ concerns. Thank you again for your prompt attention and advice regarding our submission. We hope the current version of the manuscript is suitable for publication in the International Journal of Environmental Research and Public. The changes in the text are summarized below:

Review left by reviewer #1

The authors present analyses which are apparently supplementary analyses based on a health and nutritional survey study, the Korean equivalent of the well-known NHANES format. On line 79, the design is described as "cross-sectional cohort study" which seems a contradiction in itself; according to the description with new household samples being drawn in successive years, the design would rather correspond to repeated cross-sections.

  • Thank you for your comments, which have enhanced the quality of our article. In response to the comments of the reviewers, in the 1st paragraph of the Discussion section and the 2nd paragraph of the Materials and Methods section, the expression of cross-sectional cohort study” was corrected to repeated cross-sections according to the reviewer's comments.

Statistical analysis of the outcome variables was - appropriately - by multiple logistic regression (as described in the methods section). However, in the abstract, the authors write "Using linear regression analysis, we calculated the adjusted odds ratio (OR) for each disease" which is wrong.

  • Thank you for noting this error. We agree with the opinion of the reviewer. We have corrected the statistical analysis methods from “linear regression” to multiple logistic regression in the Abstract of the revised manuscript.

The actual research question is whether atopic diseases have changed in frequency during the covid-19 epidemic; results are based on medical records which have been retrieved for the sampled individuals, comparing the years 2019 and 2020, the latter being the "Covid year". The target outcome was based on a self-reported doctor's diagnosis, not symptom-based.

As result, the authors report non-significant variation of the OR to be diagnosed for the target outcome diseases in 2020, compared to 2019 as reference year. Thereby, their initial hypothesis "We expected a decreased incidence of allergic diseases during the COVID-19 pandemic due to ... changes in lifestyle and personal habits such as universal mask-wearing and social distancing" could not be confirmed.

  • Thank you for your comments. We agree with the reviewer’s opinion. We described in the last paragraph of the Discussion section of the revised manuscript the inherent weakness of the database regarding the self-reported doctor diagnoses, that are not symptom-based as follows:

This study evaluated whether the frequency of allergic diseases changed according to changes in personal lifestyle and habits, such as wearing a universal mask during the COVID-19 pandemic. The target outcomes were based on self-reported doctor's diagnoses, not symptom-based. There will inevitably be missed diagnoses. Moreover, our results could not fully demonstrate an association between the severity of allergic diseases and lifestyle changes during the COVID-19 pandemic. This was due to the lack of information regarding the inherent weaknesses of the database.

This result could be regarded as moderately interesting if, and only if, the above hypothesis would appear plausible. Of course it can always be claimed that there is "insufficient information concerning the influence of the COVID-19 pandemic on the prevalence or severity of allergic diseases of asthma, atopic dermatitis, and allergic rhinitis" as written in the introduction. Nevertheless, such lack is not a sufficient motivation for the analyses presented. Unfortunately, the authors made no attempt to relate to current concepts of the aetiology of atopic allergies as related to lifestyle factors (e.g. nutrition, housing, early exposure to a "rural" microbiome and others) - else they would perhaps have been aware of the limited sense of their hypothesis.

  • Thank you for your comments. We agree with the reviewer’s opinion. We tried to further investigate the data related to the etiology of allergic diseases, but due to the inherent weakness of the database, we are unable to examine them due to the limited information on the timing of exposure to the symbiotic microbiota, nutrition, and housing. As suggested by the reviewer, the lack of information is accounted for in the Discussion section of the revised manuscript as follows:

The lack of information on the etiology of allergic diseases such as the timing of exposure to symbiotic microbiota, nutrition, and housing characteristics may limit our ability to support our hypothesis [50-52]. Further studies are needed to confirm the presence of these factors on prevalence.

Reviewer 2 Report

Dear Authors, 

Congratulations on Yous study! The study is very intersting, well designed, however there are some technical flaws of the manuscript I  wish you would correct. 

Here are my comments:

-in the abstract and anytime you show ORs- please underline they are insignificant- it is obvious, but if one misses the p value, there might be some false conclusions;

-in the introduction section: please add definitely more info on any of the diseases you analysed;

- also, I would strongly suggest to give more details on the rationale for taking such a study- why should be a decrese/increase in allergic diseases expected? Comment on the literature in the topic as well, for example- has anyone performed similar comparisons?, or how hygienic measures influence frequency of allergic diseases

Discussion section:

- l. 200-215 are in fact introduction, not a discussion, move or remove them;

- I do not see a direct relationship with RSV, influenza and parainfluenza in this case- of course, preventive measures influenced on the morbidity of infectious diseases, but it is not related to the topic of the study. What is more, such a comparison indirectly suggests that allergic diseases are as transmissible as infectious diseases;

- l. 229-230- you need to discuss this issue in-depth

- l. 241-244- I suppose this is the main explanation- please discuss it thouroughly

- discuss the first cases of COVID-19 in Korea- for the study you chose 2020, which was in part free of COVID (or we thought so), nevertheless you need to give the exact date when all the restrictions started and maybe use some transitional period (for example first quarter of 2020). I think that in this case, much better comparisons would be seen between 2019 and 2021- is it possible to make such an analysis?

Regarding ethical approval, I understand that the IRB stated that the study does not require its approval? If so, pleases do put it down clearly. In any other case, like  any other body or group of exxperts or authors deciding on it, the statement by IRB or another Ethics Committee is required. I suggest it, since the analysed data included some sensitive data (like personal income, educational level) that need to be carefully handled. 

Best regards

Author Response

21 September 2022

Dear Editor,

We would like to thank the reviewers for their constructive criticism, which has helped us improve the manuscript. We have carefully and thoroughly addressed all the reviewers’ concerns. Thank you again for your prompt attention and advice regarding our submission. We hope the current version of the manuscript is suitable for publication in the International Journal of Environmental Research and Public. The changes in the text are summarized below:

All changes made to the manuscript have been written in blue.

Review left by reviewer #2

Dear Authors,

Congratulations on Yous study! The study is very intersting, well designed, however there are some technical flaws of the manuscript I wish you would correct.

Here are my comments:

-in the abstract and anytime you show ORs- please underline they are insignificant- it is obvious, but if one misses the p value, there might be some false conclusions;

  • Thank you for your comment. As the reviewer noted, we removed the expression of insignificant odds ratios and added the p-values of the analyzed results as follows to aid understanding of the outcomes in the Abstract of the revised manuscript:

There were no statistically significant differences between the incidence of doctor-diagnosed and current allergic diseases in 2019 and 2020 (asthma, p = 0.667 and p = 0.268; atopic dermatitis, p = 0.268 and p = 0.973; allergic rhinitis, p = 0.691 and p = 0.942, respectively), and subgroup analysis showed consistent results.

-in the introduction section: please add definitely more info on any of the diseases you analysed;

  • Thank you for your comments. As suggested by the reviewer, information about allergic diseases including asthma, atopic dermatitis, and allergic rhinitis, has been added to the Introduction section of the revised manuscript as follows:

Atopic diseases tend to be an exaggerated immunoglobulin E-mediated immune response in response to the foreign allergen [5]. Patients with atopic traits usually present with one or more symptoms of the following disorders: asthma, atopic derma-titis, and allergic rhinitis. In recent decades, the prevalence of allergic diseases has been steadily increasing, and it currently affects about 20% of the population in devel-oped countries [6, 7]

- also, I would strongly suggest to give more details on the rationale for taking such a study- why should be a decrese/increase in allergic diseases expected? Comment on the literature in the topic as well, for example- has anyone performed similar comparisons?, or how hygienic measures influence frequency of allergic diseases

  • Thank you for your comments. We added the information supporting our hypothesis. We made corrections and additions along with the comments of the reviewers in the Discussion section of the revised manuscript below as follows:

During the COVID-19 pandemic, non-pharmacological measures such as wearing a mask, washing hands frequently, and improving indoor ventilation improve hygiene, and self-care including high medication adherence rates [28, 29]. In accordance with hygiene awareness, non-pharmacological interventions reduced exposure to outdoor exposure to pollen allergens and air pollution played a protective role, and alleviated allergic reactions. Consequently, the average daily asthma emergency treatment rate decreased by 21 to 76% compared to before the COVID-19 pandemic [30-32], and the number of children hospitalized for asthma decreased [29, 33, 34]. Reducing exposure to common respiratory viruses, which is the major triggers for asthma exacerbation, through non-pharmacological interventions and social distancing may be the most plausible explanation, according to some studies [28, 31].

We expected an overall decrease in the incidence of allergic diseases, such as reduced asthma exacerbations, during the COVID-19 pandemic. Interestingly, the results differed from what was initially expected. The trend of allergen positivity has changed following non-pharmaceutical interventions in the spread of SARS-CoV-2 infection during the COVID-19 pandemic [31].

Discussion section:

- l. 200-215 are in fact introduction, not a discussion, move or remove them;

  • Thank you for your comments. According to the reviewer’s opinion, the contents of lines 200-215 were deleted and some contents (1st and 2nd sentences) were moved to the Introduction section of the revised manuscript.

- I do not see a direct relationship with RSV, influenza and parainfluenza in this case- of course, preventive measures influenced on the morbidity of infectious diseases, but it is not related to the topic of the study. What is more, such a comparison indirectly suggests that allergic diseases are as transmissible as infectious diseases;

  • Thank you for your comments. We agree with the reviewer’s opinion that allergic diseases including asthma, atopic dermatitis, and allergic rhinitis are not transmissible infectious diseases such as viral infection diseases, but diseases caused by sensitized and provoked by inhaled allergens. According to the reviewer’s opinion, the contents in lines 200-215 of the Discussion section were totally removed.

- l. 229-230- you need to discuss this issue in-depth

  • Thank you for your comments. We added the supporting information from this sentence for better understanding in the Discussion section of the revised manuscript as follows:

This includes changing caregiver behavior for patients who are more likely to avoid visiting the hospital because of fear of transmitting SARS-CoV-2 and stopping health care [27, 28].

During the COVID-19 pandemic, non-pharmacological measures such as wearing a mask, washing hands frequently, and improving indoor ventilation improve hygiene, and self-care including high medication adherence rates [28, 29].

- l. 241-244- I suppose this is the main explanation- please discuss it thouroughly

  • Thank you for your comments. We agreed with the review's suggestion. We have described increased exposure to indoor allergens, but have been limited to supporting a main explanation. Accordingly, the contents of the Discussion section of the revised manuscript have been revised and added as follows:

In accordance with hygiene awareness, non-pharmacological interventions reduced exposure to outdoor exposure to pollen allergens and air pollution played a protective role, and alleviated allergic reactions. Consequently, the average daily asthma emergency treatment rate decreased by 21 to 76% compared to before the COVID-19 pandemic [30-32], and the number of children hospitalized for asthma decreased [29, 33, 34]. Reducing exposure to common respiratory viruses, which is the major triggers for asthma exacerbation, through non-pharmacological interventions and social distancing may be the most plausible explanation, according to some studies [28, 31]. The trend of allergen positivity has changed following non-pharmaceutical interventions in the spread of SARS-CoV-2 infection during the COVID-19 pandemic [31].

- discuss the first cases of COVID-19 in Korea- for the study you chose 2020, which was in part free of COVID (or we thought so), nevertheless you need to give the exact date when all the restrictions started and maybe use some transitional period (for example first quarter of 2020). I think that in this case, much better comparisons would be seen between 2019 and 2021- is it possible to make such an analysis?

  • Thank you for your comments. We agree with the reviewer’s opinion that 2020 includes the transitional period for social distancing. Unfortunately, we have to use only the limited data of KNHANES provided. We, therefore, added the limitation related to the study period of the present study in the last paragraph of the Discussion section as follows:

The spread of SARS-CoV-2 in the first quarter of 2020, has made mask-wearing compulsory in most areas, and many businesses have switched to telecommuting and postponed school attendance. The mandatory use of masks at the government level, which gives penalties and pays a fine for not wearing a mask, began on October 13, 2020. Although there may be a transitional period in social distancing, including mask-wearing, there was a limit to setting the year for comparative analysis because we had to analyze using only the limited data of KNHANES provided. Further studies are needed to confirm changes in allergic disease prevalence through comparisons from different years.

Regarding ethical approval, I understand that the IRB stated that the study does not require its approval? If so, pleases do put it down clearly. In any other case, like any other body or group of exxperts or authors deciding on it, the statement by IRB or another Ethics Committee is required. I suggest it, since the analysed data included some sensitive data (like personal income, educational level) that need to be carefully handled.

  • Thank you for comments. In this study, we used the Korean National Health and Nutrition Examination Survey (KNHANES) conducted by the Korean government. According to paragraph 2, Bioethics and Safety Act and subparagraph 1 of paragraph 1 of the Enforcement Rule of the Bioethics and Safety Act 2, the Institutional Review Board (IRB) approval was waived. For the research conducted directly or entrusted by the state or local governments for screening and evaluation of public welfare or service programs, research is not included in human subjects and is excluded from deliberation of the IRB. We used the KNHANES data for public welfare for government COVID-19 control that does not require deliberation by the IRB approval in accordance with the above-mentioned rule. This content has been added in the Institutional Review Board Statement section in the revised manuscript, to clarify the statement so that readers can understand as follows:

For the research conducted directly or entrusted by the state or local governments for screening and evaluation of public welfare or service programs, research is not included in human subjects and is excluded from deliberation of the IRB approval. We used the Korean National Health and Nutrition Examination Survey (KNHANES) data for public welfare for government COVID-19 control that does not require deliberation by the IRB approval in accordance with the above-mentioned rule.

Round 2

Reviewer 1 Report

The authors did not really take up the points raised. Hence, I cannot recommend the present article for publication.